# The Influence of Delayed Sealing and Repeated Air Ingress during the Storage of Maize Silage on Fermentation Patterns, Yeast Development and Aerobic Stability

**Kirsten Weiß** [1,*] , **Bärbel Kroschewski** [1] **and Horst Uwe Auerbach** [2]

1 Faculty of Life Sciences, Albrecht Daniel Thaer-Institute for Agricultural and Horticultural Siences, Humboldt-Universität zu Berlin, Invalidenstrasse 42, 10115 Berlin, Germany; b.kroschewski@agar.hu-berlin.de

2 International Silage Consultancy (ISC), Martha-Brautzsch-Straße 13, 06108 Halle, Germany; horst.uwe.auerbach@outlook.de

* Correspondence: kirsten.weiss@agrar.hu-berlin.de; Tel.: +49-(0)30-2093-46315; Fax: +49-(0)303-2093-8909

**Abstract:** This study investigates the effects of delayed sealing and repeated air ingress on the formation of primary fermentation products and other volatile organic compounds (VOC), the development of yeasts and the aerobic stability (ASTA) of maize (26.8% dry matter, DM). After packing, the silos were sealed either promptly or with a delay of 24 h, with repeated air ingress after 27, 55 and 135 days of storage. Losses of DM, fermentation pattern, including VOC, yeast numbers and aerobic stability, were determined 6 times during storage for 142 days. Yeast numbers markedly increased during the first three fermentation days, with the effect being much stronger in silage sealed with a delay than in promptly sealed silage ($\log_{10}$ cfu/g FM 7.27 vs. 5.88, $p < 0.002$). Simultaneously, the concentrations of ethanol and ethyl esters and DM losses increased. The DM losses were closely correlated with the total concentrations of alcohols and acetic acid (delay: $R^2 = 0.71$, $p < 0.001$; prompt: $R^2 = 0.91$, $p < 0.001$, respectively). The repeated air ingress for 24 h during storage after completion of the main fermentation phase had only a minor effect on fermentation pattern, VOC formation and DM losses. The relationship between the counts of total yeasts and lactate-assimilating yeasts (LAY) was very strong ($R^2 = 0.995$, $p < 0.001$), and LAY numbers were shown to be largely responsible for aerobic instability ($R^2 = 0.752$, $p < 0.001$). This trial proved the detrimental effects of air on silage fermentation with delayed sealing to be much more deleterious than repeated short-term air ingress after about one month of storage.

**Keywords:** ethanol; volatile organic compounds; maize; silage; yeasts

## 1. Introduction

The process of the anaerobic preservation of green forage is most strongly affected by the influence of atmospheric oxygen, leading not only to losses in dry matter (DM), energy and valuable feed ingredients [1], but also to the formation of volatile organic compounds that can negatively affect feed intake and animal health [2,3] as well as milk production.

Yeasts are mainly promoted by oxygen, and the more lactate-assimilating yeasts are present, the more rapid silage starts to spoil and the greater the extent of deterioration [4–7]. Under anaerobic conditions, many species ferment sugars, such as glucose, maltose and sucrose [8]. Yeasts are not only capable of producing alcohols, e.g., ethanol and propanol, which are associated with high DM losses [1,9–11], but they can also form other volatile organic substances, such as aldehydes and esters [12,13]. The presence of elevated yeast counts is generally considered a risk factor for silage spoilage after opening due to heating processes. As the filling of silos usually takes more than one day until sealing, especially on large commercial cattle farms, the effect of air may persist for a longer period of time, enabling detrimental aerobic microorganisms to remain metabolically active. The results of

studies available in the literature on the influence of air before sealing and during ensiling on yeast development and fermentation patterns have not been consistent, which may be caused by differences in aeration [14–18]. The present study investigates the effects of delayed sealing and repeated air ingress during storage on the development of yeasts and the formation of fermentation products. The focus is placed on the relationships with regard to the occurrence of alcohols and the formation of esters in connection with dry matter losses and aerobic stability during the course of fermentation.

## 2. Materials and Methods

### 2.1. Experimental Design

A 6 ha plot of half-bog soil (hypereutric chernic gleysol) belonging to a dairy farm in Trebbin (52°19′55131″; 13°24′46269″) was used for the cultivation of the late-maturing (FAO number: 320) forage maize hybrid PR38Y34 (Pioneer Hi-Bred Northern Europe Service Division GmbH, Buxtehude, Germany). The maize was planted on 26 April 2015 at a density of 83,000 plants/ha and received a total of 120 kg nitrogen per ha exclusively from cattle slurry. The forage was harvested at the early dough stage on 8 September 2015 by a Krone Big X chopper (KRONE GmbH & Co. KG, Spelle, Germany), which was set at a theoretical chop length of 10 mm. Chopped maize was thoroughly mixed and thereafter samples were taken to determine chemical composition and fungal counts (Table 1).

**Table 1.** Chemical composition and microbial counts of fresh maize before silo filling (means in g/kg DM, unless stated otherwise).

| Parameter | n | Mean | Range |
|---|---|---|---|
| Dry matter (g/kg) | 3 | 26.8 | 26.6–27.0 |
| Crude protein | 2 | 79.0 | 77.7–80.3 |
| Ether extract | 2 | 25.6 | 24.7–26.4 |
| Crude fibre | 2 | 213 | 21.2–21.4 |
| Crude ash | 2 | 45.9 | 45.4–46.5 |
| NDFom * | 2 | 453 | 44.9–45.7 |
| ADFom # | 2 | 241 | 24.1–24.1 |
| ADL + | 2 | 31.6 | 31.4–31.9 |
| Water-soluble carbohydrates | 2 | 158 | 157–160 |
| Buffering capacity † | 2 | 27.4 | 26.6–28.2 |
| Nitrate | 2 | 1.6 | 1.6–1.6 |
| Total yeast count ($\log_{10}$ cfu/g) | 6 | 4.4 | 4.3–4.5 |
| Mould count ($\log_{10}$ cfu/g) | 6 | 5.5 | 5.3–5.8 |

* NDFom = neutral detergent fibre, expressed exclusive of residual ash; # ADFom = acid detergent fibre, expressed exclusive of residual ash; + ADL = acid detergent lignin; † g lactic acid/kg DM.

After manual packing into 1.5 L glass jars (Weck, Öfingen, Germany) at a density of $198 \pm 2$ kg DM/m$^3$, the experimental silos were either immediately sealed (prompt) or with a delay of 24 h (delay). Likewise, half of the silos were stored without or with air ingress (Ai). On days 27, 55 and 135 of storage, the Ai jars were opened for 24 h by removing the rubber stopper from the holes (diameter 6 mm) in the jar 6 cm above the bottom and the lid to enable gas exchange. The above-described aeration protocol was used to reflect long silo filling periods caused by bad weather conditions, or simply due to the fact that on large cattle farms normal-sized silos cannot be filled in one day. Repeated air ingress during storage aimed at simulating the effects of plastic film damages by rodents or birds, which were recognized early and sealed by applying self-adhesive tape. For each of the tested treatments prompt and delay up to day 27 of storage, there was no air ingress in all treatments with six replicates per treatment. The influence of the air during the storage were tested on three replicates in the treatments prompt, prompt_Ai, delay, and delay_Ai after 34, 62 and 142 days of storage, respectively. A total of 72 experimental silages were produced, which were kept at 20–22 °C throughout the entire storage period.

## 2.2. Determination of Dry Matter (DM)

The content of DM in fresh forage and silage was determined at 60 °C until a constant weight was reached, followed by drying at 105 °C for 3 h [19]. According to Weissbach and Strubelt [20], the silage DM concentration was corrected for the loss of volatiles during drying.

## 2.3. Chemical Analyses

All samples were kept in a freezer at −18 °C until analysis. The fresh forage was freeze dried and ground over a sieve (1 mm) prior to analysis (Gamma 1-16 LSC, Martin Christ, Osterode, Germany).

Nutrients were determined according to VDLUFA [21], the official German methods for feed evaluation. Water-soluble carbohydrates (WSC) were analysed by the anthrone method [22] using a continuous flow analyser (CFA, Scan++, Skalar Analytical, Breda, Netherlands). The determination of organic acids, alcohols and esters was performed in aqueous silage extracts, which were prepared by blending 50 g of the frozen material with 200 mL distilled water and 1 mL toluene. The extracts were kept at 4 °C overnight and then filtered through an MN 615 filter paper (Machery-Nagel, Düren, Germany), followed by micro-filtration (Minisart RC, 0.45 μm pore size, Sartorius, Göttingen, Germany). The pH was analysed potentiometrically by using a calibrated pH electrode (Deutsche METHROM GmbH Co. KG, Filderstadt, Germany). According to Weiss and Kaiser [23], lactic acid was detected by high performance liquid chromatography (HPLC) using refractive index (RI) detection (LC-20 AB, Shimadzu Deutschland, Duisburg, Germany). Volatile organic acids (acetic, propionic, butyric acids) and alcohols were determined by gas chromatography (GC) using a free fatty acid phase (Permabond FFAP 0.25 μm, Macherey-Nagel, Düren, Germany) column and flame ionization detector (GC-2010, Shimadzu, Deutschland, Duisburg, Germany), as described by Weiss and Sommer [24]. The limit of detection for each parameter was 0.01% of fresh matter. Other volatile organic compounds (VOC), e.g., ethyl esters of lactic and acetic acids, methanol, n-propanol, 2-butanol, were also determined by GC with FID using a 0.25 μm Optima Wax column (Macherey-Nagel, Düren, Germany). The extracts were supplemented with the internal standard 2-methyl pentanol. The detailed description of the method, including its precision parameters, was published by Weiss and Sommer [25]. The detection limit of each VOC was determined to be 3 mg/L, or 0.001% of fresh matter.

The level of effective (undissociated) acetic acid ($AA_{eff}$) to suppress yeast activity was calculated as follows [26] $AA_{eff} = AA * H^+/(H^+ + K_D)$, where $K_D$ is the dissociation constant of acetic acid and $H^+$ is the concentration of hydrogen ions.

## 2.4. Aerobic Stability Measurement

The aerobic stability of the silages was measured over 14 days by using the temperature method, according to Honig [27]. Silage was loosely filled and placed in plastic tubes (100 mm diameter, 250 mm heights) with data loggers (Tinytag Talk 2, Gemini, Chichester, UK) inserted into the geometric centre and stored at 20 °C. The data loggers recorded silage and room temperature at 2 h intervals. Each plastic pipe was stored in an insulating polystyrene box, allowing free air circulation. Silages were considered aerobically unstable once the silage temperature had reached 2 °C above ambient.

## 2.5. Microbiological Analyses

Fresh maize and maize silage were kept cool until analysis, which was carried out within 4 h after sampling. Microbiological analyses were performed by the accredited laboratory BECIT GmbH, Bitterfeld-Wolfen, Germany. After the preparation of serial sample dilutions with peptone water broth (1 g/L), the counts of yeast and moulds were enumerated after spread plating on yeast-extract-dextrose-chloramphenicol agar and incubated for 3 to 5 d at 25 °C (ISO 21527). The count of lactate-assimilating yeasts (LAY) was determined on yeast-nitrogen-base agar supplemented with lactic acid as the sole carbon source [4].

*2.6. Statistical Analysis*

The experimental data were analysed separately for each storage time in a framework of a fixed effects model using sealing time (ST) and air ingress (AI) as experimental factors. Air ingress started from day 28, so for day = 3, 7, and 16 only the effect of ST had to be analyzed in a one-factorial model [28]:

$$yik = \mu + ti + eik \qquad (1)$$

where *yik* is the observed value of the *k*th replication from sealing time *i*; $\mu$ is the population mean; *ti* is the fixed effect of sealing time *i*; and *eik* is the random residual effect of the *i*th treatment and *k*th observation. For day = 34, 62, and 142, we used a two-factorial model:

$$yijk = \mu + ti + aj + (ta)ij + eijk \qquad (2)$$

where *yijk* is the observed value of the *k*th replication from sealing time *i* and air ingress level *j*; $\mu$ is the population mean; *ti* the fixed effect of sealing time *i*; *aj* the is fixed effect of air ingress *j*; (*ta*)*ij* is the fixed interaction effect of both treatment factors; and *eijk* is the random residual effect of the *ij*th treatment and *k*th observation.

For most traits, the normality of observations could be assumed. Main effects (model 1, 2) and interaction effects (model 2) were tested by the global *F*-test and pairwise comparisons between least square means were made by Tukey's test procedure, considering interactions between both treatment factors in model (2). Due to the small sample size, variance homogeneity could not be proven, and inferences are based on a common residual variance.

The counts of lactate-assimilating yeasts were $\log_{10}$-transformed before analysis. Values below the detection limit of 100 cfu/g of fresh matter were set at the detection limit ($\log_{10}$ 2.0).

For the yeast count, aerobic stability, 2-butanol, and n-propanol, the assumption of normally distributed data could not be supported so that a rank procedure with the ANOVA-type statistical model for the global F test and pairwise comparisons among the rank means (Proc MIXED) was alternatively used. The relationships among silage variables were analyzed using linear or quasi-linear regression models. The model fit was evaluated by the coefficient of determination ($R^2$) adjusted for degrees of freedom.

Statistical analysis was performed using MIXED and REG procedures by SAS 9.4 software (SAS Institute Inc., Cary, NC, USA). Statistical significance was declared at $p \leq 0.05$.

## 3. Results

*3.1. Effects of Delayed Sealing and Repeated Air Ingress on Fermentation Patterns, Water Soluble Carbohydrate Concentration (WSC) and Dry Matter (DM) Losses*

3.1.1. Effects of Sealing Time during the Early Phase of Fermentation until Day 16

Sealing time affected the production of lactic acid in the first seven days of fermentation, which was reflected by lower concentrations, and higher pH caused by delayed sealing (Table 2). Acetic acid was rapidly formed during the first seven days with concentrations exceeding 10 g/kg DM, whereas delayed sealing increased acetic acid formation. After 16 days of anaerobic storage, the pH was significantly higher in the delayed ensiled treatments. However, there were no longer any differences between lactic acid and acetic acid contents. In turn, DM losses were consistently lower in promptly sealed silages.

The increase in ethanol content, which was higher in silages sealed with a delay, was accompanied by a marked decrease in WSC concentration to about one-third found in fresh forage at ensiling (158 g/kg DM), in silage sealed promptly (48.9 g/kg DM), and to about a tenth in those sealed with delay (13.4 g/kg DM) (Tables 1 and 2). Ethyl esters of lactic (EL) and acetic (EA) acids were already detected on the first sampling day and their concentrations increased, with higher levels being found in silages sealed with a delay. A very rapid and intensive production of EA was observed on day 3 of storage, with concentrations reaching 736 (prompt) and 2850 mg/kg DM (delay). However, the rate of EA formation and the concentration attained after 16 days was higher than that of EL.

**Table 2.** Effects of sealing time (ST; 0 h = prompt vs. 24 h = delay) and air ingress (Ai) on fermentation characteristics and dry matter (DM) losses during the course of maize fermentation ($n$ = 6: d 3, 7, and 16; $n$ = 5: d 16, ST = Prompt; $n$ = 3: d 34, 62, and 142).

| Storage Length | ST | Ai | pH | Lactate | Acetate | Ethanol | WSC | DM Loss |
|---|---|---|---|---|---|---|---|---|
| Length (Days) | | | | (g/kg DM) | (g/kg DM) | (g/kg DM) | (g/kg DM) | (%) |
| 3 | Prompt | - | 4.02 [a] | 13.9 [b] | 10.7 [a] | 6.8 [a] | 48.9 [b] | 3.6 [a] |
| | Delay | - | 4.22 [b] | 11.0 [a] | 12.6 [b] | 19.7 [b] | 13.4 [a] | 10.5 [b] |
| SEM | | | 0.01 | 0.41 | 0.29 | 0.59 | 0.57 | 0.06 |
| Effects [†] | ST | | <0.001 | <0.001 | <0.001 | <0.001 | <0.001 | <0.001 |
| 7 | Prompt | - | 3.79 [a] | 24.2 [b] | 12.1 [a] | 17.9 [a] | 11.7 [b] | 5.0 [a] |
| | Delay | - | 4.05 [b] | 19.5 [a] | 13.2 [b] | 20.5 [b] | 7.5 [a] | 10.8 [b] |
| SEM | | | 0.01 | 0.48 | 0.34 | 0.50 | 0.21 | 0.11 |
| Effects [†] | ST | | <0.001 | <0.001 | 0.042 | 0.004 | <0.001 | <0.001 |
| 16 | Prompt | - | 3.77 [a] | 27.5 [a] | 13.7 [a] | 19.4 [a] | 7.7 [b] | 5.3 [a] |
| | Delay | - | 3.92 [b] | 26.2 [a] | 12.7 [a] | 21.1 [a] | 6.0 [a] | 11.2 [b] |
| SEM | | | 0.01; 0.01 | 0.87; 0.79 | 0.53; 0.48 | 0.72; 0.66 | 0.39; 0.35 | 0.09; 0.08 |
| Effects [†] | ST | | <0.001 | 0.286 | 0.203 | 0.113 | 0.009 | <0.001 |
| 34 | Prompt | - | 3.76 [a] | 31.6 [a] | 14.8 [a] | 20.4 [a] | 6.8 [a] | 5.3 [a] |
| | | + | 3.76 [a] | 28.8 [a] | 16.8 [b] | 21.1 [a] | 6.5 [a] | 5.8 [b] |
| | Delay | - | 3.89 [c] | 31.4 [a] | 17.4 [b] | 27.9 [b] | 6.0 [a] | 11.6 [c] |
| | | + | 3.83 [b] | 30.9 [a] | 18.3 [b] | 29.0 [b] | 6.3 [a] | 12.1 [d] |
| SEM | | | 0.004 | 0.86 | 0.43 | 0.68 | 0.25 | 0.07 |
| Effects [†] | ST | | <0.001 | 0.299 | 0.001 | <0.001 | 0.076 | <0.001 |
| | Ai | | <0.001 | 0.092 | 0.011 | 0.205 | 0.994 | <0.001 |
| | ST × Ai | | <0.001 | 0.211 | 0.231 | 0.800 | 0.254 | 0.991 |
| 62 | Prompt | - | 3.74 [a] | 34.0 [b] | 16.9 [a] | 20.7 [a] | 6.7 [a] | 5.4 [a] |
| | | + | 3.74 [a] | 28.7 [a] | 18.5 [a] | 21.5 [a] | 6.8 [a] | 6.0 [b] |
| | Delay | - | 3.83 [c] | 33.6 [b] | 20.9 [b] | 27.3 [b] | 6.1 [a] | 11.8 [c] |
| | | + | 3.79 [b] | 31.9 [ab] | 22.0 [b] | 26.7 [b] | 6.8 [a] | 12.2 [d] |
| SEM | | | 0.002 | 1.04 | 0.42 | 0.69 | 0.16 | 0.07 |
| Effects [†] | ST | | <0.001 | 0.205 | <0.001 | <0.001 | 0.138 | <0.001 |
| | Ai | | <0.001 | 0.010 | 0.014 | 0.895 | 0.028 | <0.001 |
| | ST × Ai | | <0.001 | 0.114 | 0.610 | 0.330 | 0.147 | 0.129 |
| 142 | Prompt | - | 3.86 [a] | 34.8 [b] | 25.2 [a] | 20.8 [a] | 6.6 [ab] | 6.0 [a] |
| | | + | 3.87 [a] | 28.0 [a] | 28.4 [a] | 21.0 [a] | 7.5 [b] | 6.9 [a] |
| | Delay | - | 3.88 [a] | 33.4 [b] | 27.9 [a] | 32.3 [b] | 5.8 [a] | 11.9 [b] |
| | | + | 3.88 [a] | 31.8 [ab] | 28.7 [a] | 31.6 [b] | 6.2 [a] | 12.8 [b] |
| SEM | | | 0.01 | 1.03 | 1.14 | 0.85 | 0.22 | 0.21 |
| Effects [†] | ST | | 0.120 | 0.293 | 0.233 | <0.001 | 0.002 | <0.001 |
| | Ai | | 0.737 | 0.003 | 0.113 | 0.782 | 0.017 | 0.003 |
| | ST × Ai | | 0.737 | 0.037 | 0.349 | 0.601 | 0.236 | 0.958 |

- no air ingress during storage; + air ingress for 24 h during storage on day 27, 55, 135 d of storage; [†] $p$ values of global *F*-test, a–d: least-square means in columns within storage length differ if they have no letter in common ($p < 0.05$; Tukey's test); SEM based on residual variance.

### 3.1.2. Effects of Sealing Time and Air Ingress during the Later Phase of Fermentation from Day 32

Air ingress after 27, 55 and 135 days of storage resulted in a lower lactic acid content in the promptly sealed silage but increasing acetic acid concentrations in all treatments up to day 142 of storage, despite comparably low pH values and similarly low and WSC levels (Table 2). Regardless of air ingress, ethanol production continued to increase beyond day 16 of storage, exceeding 30 g/kg DM at the end of the experimental period in silage sealed with a delay. The formation of ethanol and EL followed a very similar pattern during the entire fermentation process (Tables 2 and 3), with significantly higher ethanol and ester contents in delayed silages. At the end of the storage, delayed silages without and with air ingress contained the highest level of ethanol (32.3 and 31.6 g/kg DM) and EL (508 and 503 mg/kg DM). The pattern of ethyl ester accumulation differed between EA and EL. The concentration of EA decreased after reaching a maximum (1520 mg/kg DM) on day 16 of storage in the promptly sealed silages, whereas in silages sealed with delay the highest concentrations were detected on day 34 with lower levels found thereafter until the end of storage. The highest EA level was measured on day 34 in delayed silage with and without

repeated air ingress (3895 and 4556 mg/kg DM) compared with promptly sealed silages (1127 vs. 1344 mg/kg DM, ST × Ai interaction, *p* < 0.026). In promptly sealed silages, the EA formation was more reduced on sampling day 34, 62 and 142 (*p* < 0.001). The formation of EL peaked on day 62 in promptly sealed silage and on day 34 in silages sealed with a delay and remained unaffected by storage length thereafter. Repeated air ingress during later stages of fermentation had no influence on EA and EL contents in promptly sealed silages, with the exception of EA in silages sealed with a delay and sampled on day 62.

**Table 3.** Effects of sealing time (ST; 0 h = prompt vs. 24 h = delay) and air ingress (Ai) on volatile organic compounds (VOC) during the course of maize fermentation (*n* = 6: d 3, 7, and 16; *n* = 5: d 16, ST = Prompt; *n* = 3: d 34, 62, and 142).

| Storage | ST | Ai | Methanol | n-Propanol | 2-Butanol | Ethyl Acetate | Ethyl Lactate |
|---|---|---|---|---|---|---|---|
| Length (Days) | | | (mg/kg DM) | | | | |
| 3 | Prompt | - | 141 [a] | 0 [a] | 0 [a] | 736 [a] | 27 [a] |
| | Delay | - | 107 [a] | 4 [a] | 0 [a] | 2850 [b] | 76 [b] |
| SEM | | | 24.5 | 0;4.3 | 0 | 126.9 | 11.6 |
| Effects [†] | ST | | 0.353 | 0.363 | | <0.001 | 0.013 |
| 7 | Prompt | - | 107 [a] | 0 [a] | 0 [a] | 815 [a] | 157 [a] |
| | Delay | - | 96 [a] | 0 [a] | 0 [a] | 2689 [b] | 217 [b] |
| SEM | | | 17.6 | 0 | 0 | 108.6 | 10.4 |
| Effects [†] | ST | | 0.666 | | | <0.001 | 0.002 |
| 16 | Prompt | - | 165 [b] | 0 [a] | 0 [a] | 1520 [a] | 217 [a] |
| | Delay | - | 93 [a] | 8 [a] | 151 [b] | 3115 [b] | 289 [b] |
| SEM | | | 23.3; 21.2 | 0;8.0 | 0; 32.9 | 266.8; 243.6 | 10.0; 9.1 |
| Effects [†] | ST | | 0.048 | 0.363 | <0.001 | 0.002 | <0.001 |
| 34 | Prompt | - | 170 [a] | 0 [a] | 0 [a] | 1127 [a] | 242 [a] |
| | | + | 240 [a] | 0 [a] | 0 [a] | 1344 [a] | 246 [a] |
| | Delay | - | 219 [a] | 0 [a] | 320 [b] | 4556 [b] | 405 [b] |
| | | + | 267 [a] | 0 [a] | 125 [ab] | 3895 [b] | 406 [b] |
| SEM | | | 41.4 | 0 | 0–15.3 | 161.2 | 5.6 |
| Effects [†] | ST | | 0.387 | | <0.001 | <0.001 | <0.001 |
| | Ai | | 0.192 | | 0.021 | 0.205 | 0.6850.990 |
| | ST × Ai | | 0.796 | | 0.021 | 0.026 | 0.803 |
| 62 | Prompt | - | 222 [a] | 0 [a] | 0 [a] | 732 [a] | 275 [a] |
| | | + | 269 [b] | 12 [ab] | 0 [a] | 764 [a] | 284 [a] |
| | Delay | - | 277 [b] | 159 [b] | 352 [b] | 3750 [c] | 401 [b] |
| | | + | 266 [b] | 0 [a] | 144 [ab] | 2678 [b] | 414 [b] |
| SEM | | | 9.1 | 0–12.4 | 0–25.1 | 181.0 | 12.8 |
| Effects [†] | ST | | 0.021 | 0.065 | <0.001 | <0.001 | <0.001 |
| | Ai | | 0.084 | 0.065 | 0.021 | 0.021 | 0.422 |
| | ST × Ai | | 0.013 | 0.022 | 0.021 | 0.016 | 0.904 |
| 142 | Prompt | - | 245 [a] | 1889 [a] | 24 [a] | 268 [a] | 334 [a] |
| | | + | 240 [a] | 2321 [a] | 33 [ab] | 284 [a] | 358 [a] |
| | Delay | - | 369 [b] | 835 [a] | 362 [b] | 1507 [b] | 508 [b] |
| | | + | 358 [ab] | 868 [a] | 231 [b] | 1374 [b] | 503 [b] |
| SEM | | | 37.6 | 30.2–1026.9 | 4.3–39.2 | 88.3 | 24.9 |
| Effects [†] | ST | | 0.012 | 0.430 | 0.001 | <0.001 | <0.001 |
| | Ai | | 0.838 | 0.786 | 0.387 | 0.524 | 0.713 |
| | ST × Ai | | 0.938 | 0.892 | 0.059 | 0.423 | 0.576 |

- no air ingress during storage; + air ingress during storage (on day 27,55,135 of storage); [†] *p*-values of global *F*-test or non-parametric global test; a–c: least-square means in columns within storage length differ if they have no letter in common (*p* < 0.05; Tukey's test or non-parametric rank test); SEM for yeast count based on individual treatments, for other variables based on residual variance.

Methanol was found from day 3, whereas 2-butanol was first detected on day 16 of storage. In silages sealed with a delay, higher contents of methanol and 2-butanol were observed from day 62 of storage regardless of air ingress during storage (Table 3), on day 142 up to 369 and 358 mg/kg DM methanol and 362 and 231 mg/kg DM 2-butanol. Propanol was formed only after day 62 of storage, without effect of sealing time and air ingress and up to mean levels of 1478 mg/kg DM.

### 3.2. Effects of Delayed Sealing and Repeated Air Ingress on Yeast Development and Aerobic Stability (ASTA)

In relation to the yeast content of 4.4 $\log_{10}$ cfu/g in the ensiled forage (Table 1), the numbers of lactate-assimilating yeasts (LAY) increased extremely during the first three days of storage (Figure 1), with the effect being much stronger in silages sealed with a delay than in promptly sealed silages ($\log_{10}$ cfu/g 7.27 vs. 5.88, $p < 0.002$). Up to day 16 of storage, LAY counts were higher in silages sealed with delay, and a decline was observed earlier (from day 3) than in promptly sealed silages (from day 7). Regardless of sealing time, air ingress on day 27 of storage resulted in a higher LAY count than in silages without air ingress ($\log_{10}$ cfu/g 3.49 vs. 2.70, $p < 0.001$). In silages sealed with a delay and air ingress, the LAY count increased from day 34 (3.02 $\log_{10}$ cfu/g) to $\log_{10}$ cfu/g 4.73 on day 62, with no further changes observed until the end of storage.

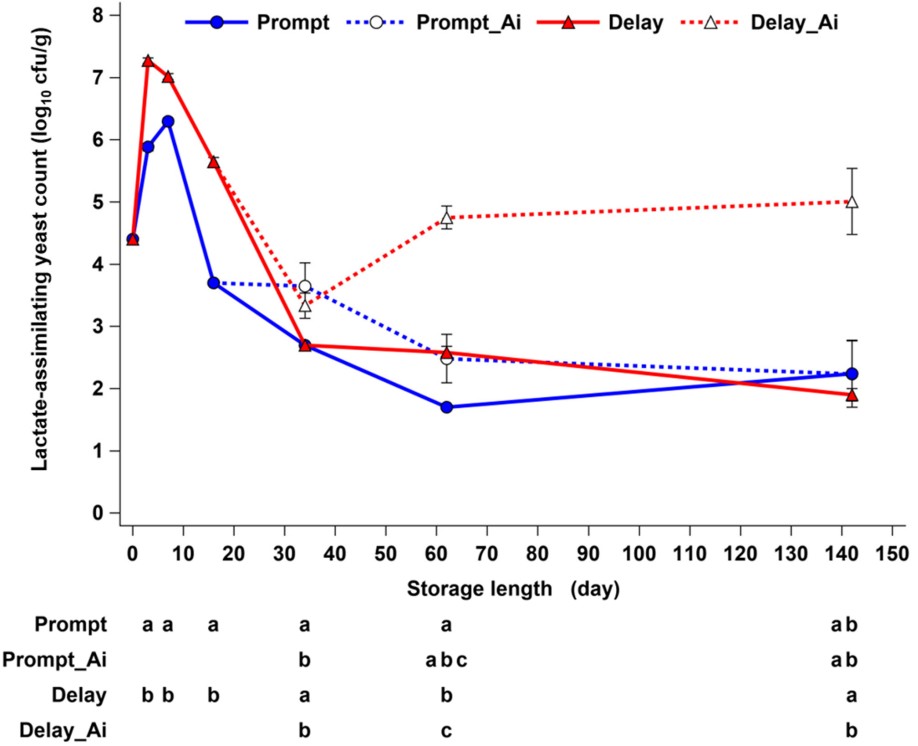

**Figure 1.** Changes in the concentration of lactate-assimilating yeasts during the fermentation of maize silage as affected by sealing time and repeated air ingress during 142 days of storage. Least square means are presented with standard error of means as error bars, significant differences within respective storage length are given below as plot table (a–c: means with no common letter differ at $p < 0.05$; pairwise comparisons by rank procedure). Prompt, promptly sealed, without air ingress during storage; Prompt_Ai, promptly sealed with air ingress after 27, 55, 135 days of storage; Delay, sealed with 24 h delay, without air ingress during storage; Delay_ Ai, sealed with 24 h delay and air ingress after 27, 55, 135 days of storage.

Based on linear regression analysis (Figure 2), total yeast count was almost exclusively composed of lactate-assimilating species ($R^2 = 0.995$; $p < 0.001$).

Despite differences between sealing time, aerobic stability (ASTA) was always below two days during the first week of storage (Figure 3). Thereafter, ASTA increased in promptly sealed silages without air until the end of fermentation. Throughout storage, silage sealed with delay and repeated air ingress showed the lowest ASTA. A strong exponential relationship was found between the count of LAY and ASTA (Figure 4, $R^2 = 0.752$, $p < 0.001$).

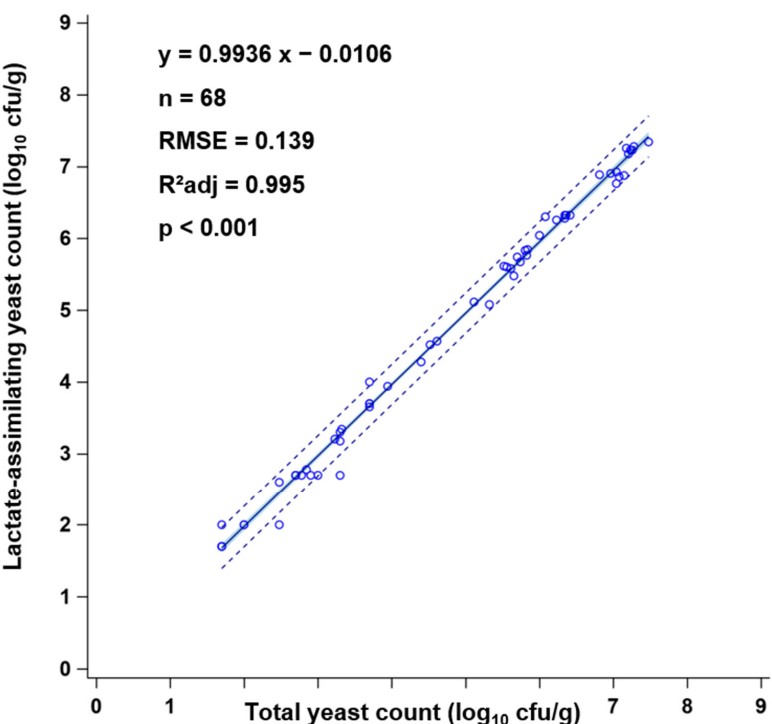

**Figure 2.** Relationship between the counts of total and lactate-assimilating yeasts in maize silages.

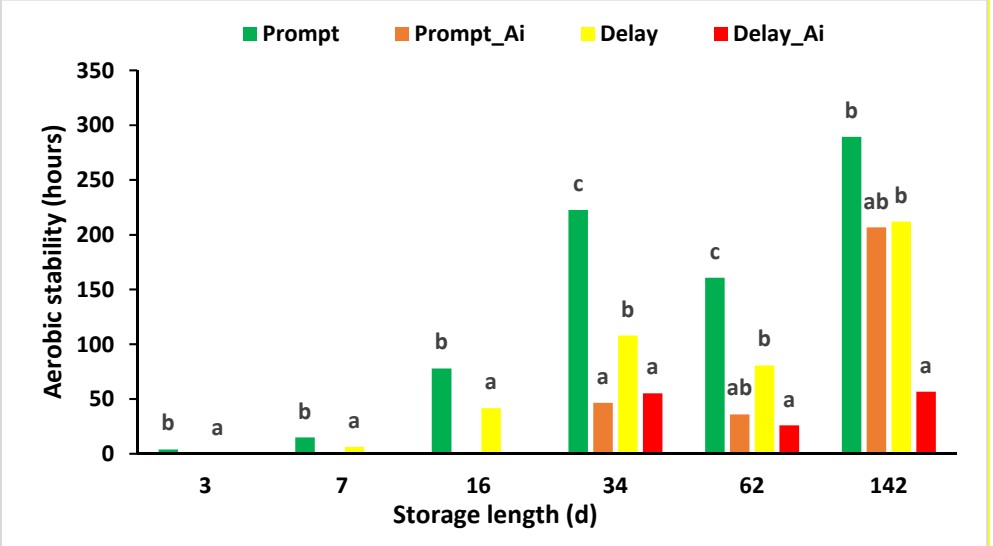

**Figure 3.** Aerobic stability during the course of fermentation of maize silage as affected by sealing time and repeated air ingress during storage of maize silage over 142 days. Least square means are presented with standard error of means as error bars, significant differences within respective storage length are given below as plot table (a–c: means with no common letter differ at $p < 0.05$; pairwise comparisons by rank procedure). Prompt, promptly sealed, without air ingress during storage; Prompt_Ai, promptly sealed with air ingress after 27, 55, 135 days of storage; Delay, sealed with 24 h delay, without air ingress during storage; Delay_ Ai, sealed with 24 h delay and air ingress after 27, 55, 135 days of storage.

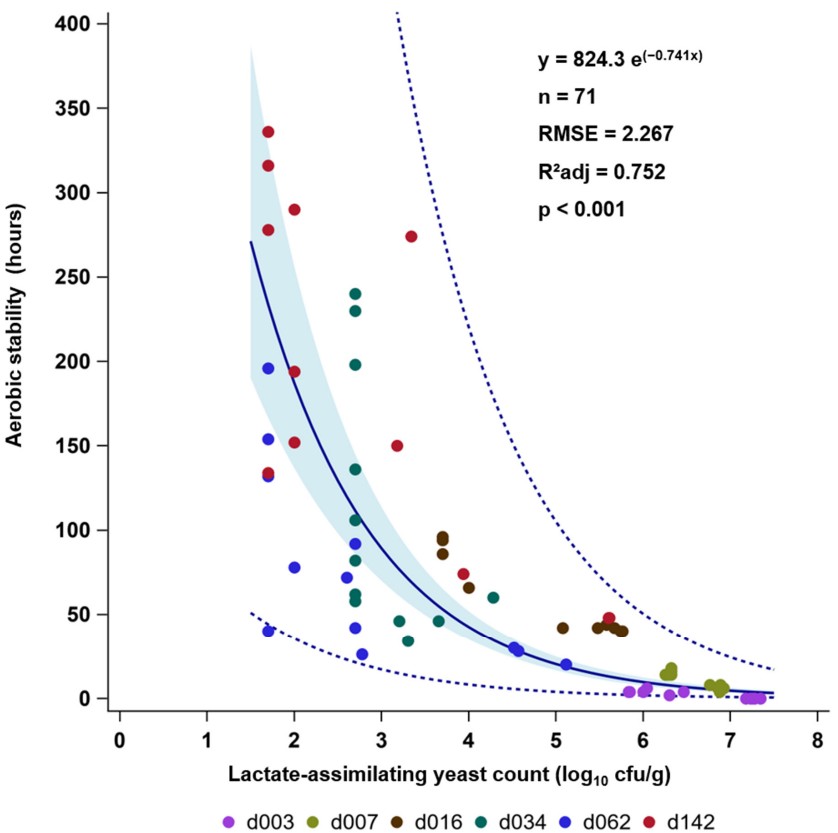

**Figure 4.** Relationship between the count of lactate-assimilating yeasts and aerobic stability of maize silage determined on day 3 (d003), day 7 (d007), day 16 (d016), day 34 (d034), day 62 (d062), and day 142 (d142) of storage.

## 4. Discussion

Repeated air ingress after 27, 55, and 135 days of storage for 24 h one week prior to silo opening hardly influenced the fermentation pattern in contrast to results of Kung et al. [17,29]. The causes of these differences cannot be adequately explained, but they may be related to a different aeration protocol and a largely different DM of 37% as opposed to 26.8% in our study, which has a strong effect on fermentation intensity [30]. The detected marginal decrease in lactate concentration from the day 62 of storage due to air ingress agrees with previous results from Bolsen et al. [31] and McEniry et al. [16]. We believe that the relatively high packing density attained in the laboratory silo (198 $\pm$ 2 kg DM/m$^3$) at this low DM content of 26.8% resulted in low air circulation and thus low oxygen content in the pore volume, thereby supporting intensive lactic acid-dominated fermentation pathways. On the contrary, a 24 h sealing delay drastically increased yeast activity already during the first three days of storage compared to prompt sealing. Furthermore, the more pronounced depletion of the initially present WSC concentration in the forage (158.5 g/kg DM) to 13.4 g/kg DM in silages sealed with delay, as opposed to 48.9 g/kg DM in promptly sealed silages, resulted in lower quantities of sugar available for lactic acid production. Jonsson and Pahlow [4], Petersson [32] and Mostagi and Wittenburg [14] have already described these effects. Even in compacted laboratory silos, both aerobic metabolic processes in the upper section of the silo and anaerobic yeast activity in the middle to lower sections of the laboratory silo may have occurred due to delayed sealing. Significantly, higher ethanol levels in the first three days confirm that most yeast species require a certain oxygen level for anaerobic ethanol formation [8,30]. Other metabolic end-products of yeast activity, e.g., propanol and butanol [33], could not be detected in the present study until day 16 of storage. Furthermore, the higher ethanol and acetic acid contents in treatments sealed with a delay compared to promptly sealed silage during the first seven days of storage may likely be attributed to the

aerobic activity of enterobacteria [29], or to oxygen-induced changes of the metabolism by lactic acid bacteria [34].

Whereas lactic acid formation after 34 days remained at the same level (average of all treatments: 30.7 g/kg DM) regardless of sealing time and aeration, the concentration of acetic acid was significantly higher in silages sealed with a delay. Due to the decrease in pH, which is equivalent to an increase in hydrogen ion concentration, in all treatments to values of 3.76 (prompt) and 3.86 (delay), a shift in the dissociation equilibrium of acetic acid in favor of the undissociated portion [26,35,36] can be assumed. All these studies demonstrated that undissociated acetic acid (effective acetic acid) has an inhibitory effect on yeasts.

The data of the present study clearly show that, in particular, the higher the concentration of effective acetic acid, the lower the count of lactate-assimilating yeasts up day 34 of storage (Figure 5, $R^2$ =0.664, *p* < 0.001). The reduction of yeast numbers in all treatments to levels below $10^4$ log cfu/g was correlated with an increase in aerobic stability from day 7 to day 16 and to day 34, respectively. The relationship between acetic acid level, here effective acetic acid, and aerobic stability confirmed results by Wolthusen [26], Danner et al. [37], Kim and Adesogan [38], and Comino et al. [39].

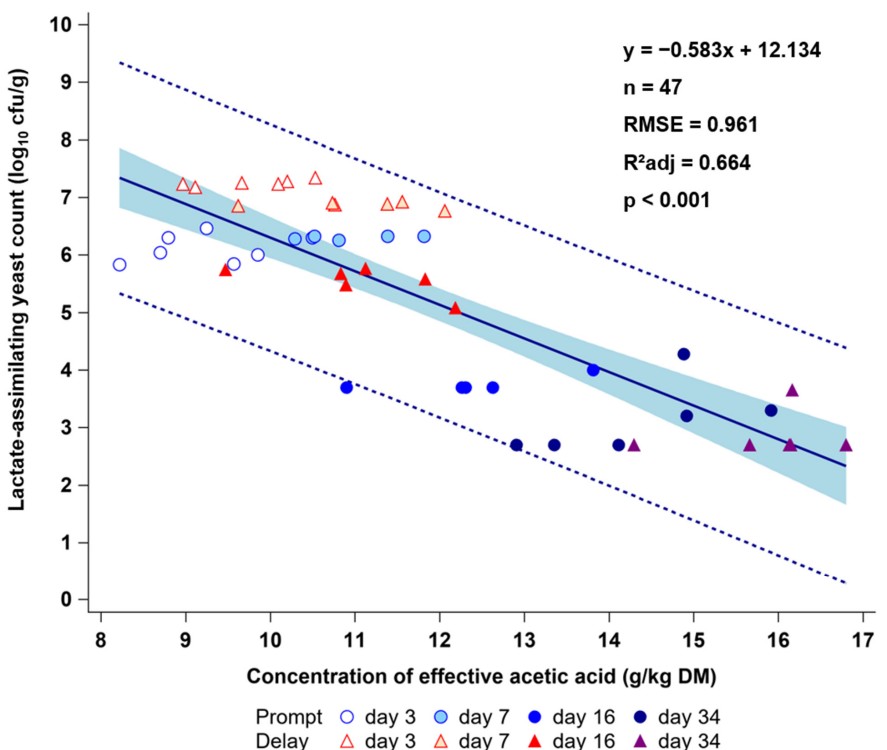

**Figure 5.** Relationship between the concentration of effective (undissociated) acetic acid and the count lactate-assimilating yeast count in maize silage during 34 days of storage.

Auerbach et al. [40] demonstrated negative relationships between yeast count and ASTA (y = 819.4 x−1.912, $R^2$ = 0.98, *p* < 0.001) with increasing acetic acid levels up to a maximum of 30 g/kg DM (y = 10.65 $e^{0.1702x}$, $R^2$ = 0.96, *p* < 0.001) in their experiments with grass and early cut rye silage [41]. The authors concluded from their experimental results regarding content of yeasts in relation to WSC (y = 739.1 $x^{−1.243}$, $R^2$ = 0.94, *p* < 0.001) that the silages with the lowest contents of water-soluble carbohydrates can have the highest ASTA. This correlation could not be supported in our experiment, as the WSC were low in all treatments as from day 16 with values below 10 g/kg DM until the end of storage. However, as from day 16 of storage, ASTA was highly variable ranging between <50 h and >200 h depending on the treatment. Wilkinson and Davis [42] also made reference to

several papers in their review that did not find a direct relationship between the level of WSC and ASTA.

The DM losses in our study were strongly correlated with intensive yeast activity, alcohol and acetic acid formation [1], in both sealing time treatments (Figure 6, $R^2 = 0.71$, $p < 0.001$; $R^2 = 0.92$, $p < 0.001$). Assuming that the intercept with the *y*-axis (sum of alcohols and acetic acid) represents the unavoidable fermentation loss due to $CO_2$ production under strictly anaerobic conditions (2–4% according to Zimmer [43]), here 2.3% in promptly sealed silage, the mean difference for those sealed with a delay is 6.7%, or a fourfold higher fermentation loss due to delayed sealing. According to McDonald et al. [1] and Rooke and Hatfield [33], DM losses are generally much higher when bacteria other than lactic acid bacteria play a dominant role in fermentation. The present results confirmed the importance of immediate air exclusion to safely prevent intensive aerobic conversion processes during the first days of storage. Brüning et al. [44] also found in their experiments with maize that a delay of sealing by four days resulted in similar fermentation losses of up to 11%. This data clearly showed that any undesired fermentation processes that generate $CO_2$ must be suppressed to maintain the highest DM recovery possible.

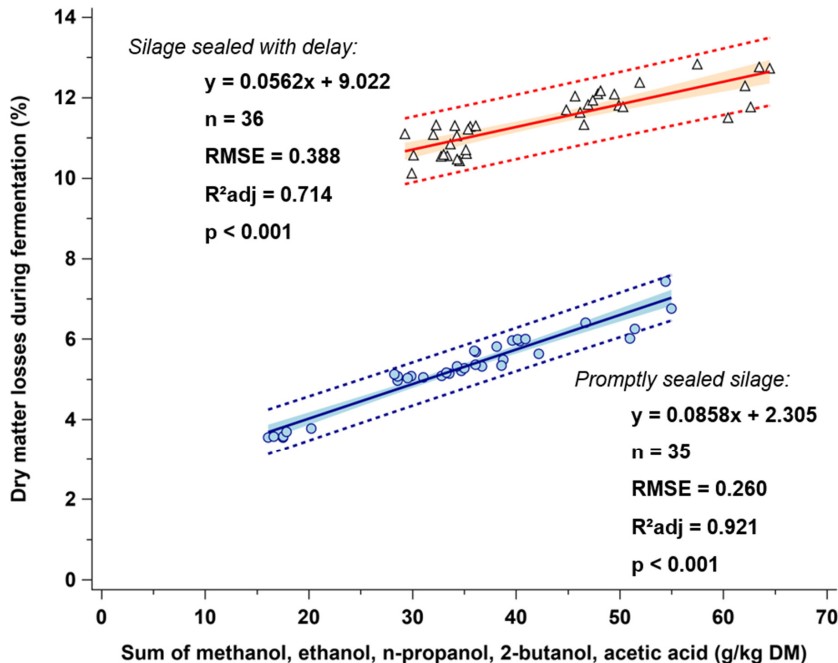

**Figure 6.** Dry matter losses as a function of the total concentration of methanol, ethanol, n-propanol, 2-butanol and acetic acid in in maize silages sealed promptly or with a delay.

In conjunction with increased yeast activity during the early stages of fermentation especially in silages sealed with a delay, significant quantities of ethyl esters were detected, of which ethyl acetate can be formed chemically and directly by yeasts [13]. On the contrary, ethyl lactate is the end-product of the reaction of ethanol and lactic acid, which is supported by our data showing an increase in the concentration of this compound with increasing levels of the reactants. While the formation of 2-butanol by yeasts occurred to a minor extent, but especially in silage sealed with a delay, the alcohol n-propanol, also formed by yeasts, occurred only at the end of storage with values up to about 1.5 g/kg DM and without formation of the ester propyl acetate, regardless of sealing time and air ingress. However, it remains unclear as to whether other n-propanol-producing microorganisms, e.g., *Lactobacillus diolivorans* [10] were active.

Methanol was detected from the start of fermentation, with levels ranging from an average of 124 mg/kg DM on day 3 to 303 mg/kg DM on day 142. According to Schink and Zeikus [45], methanol is a major product of pectin metabolism by aerobic,

facultative anaerobic, and obligate anaerobic pectinolytic bacteria, but also by yeasts and fungi that produce pectin methyl esterase [46–48]. Due to the general occurrence of pectins as polysaccharides with stabilizing and supports functions in the primary wall of plant cells, pectins are also present in maize [49]. Hafner et al. [50] pointed out that the methanol found in silages is due to the activity of plant enzymes prior to ensiling, as a product of pectin demethylation during leaf expansion and cell wall synthesis. However, in our study, an increase in methanol content was observed in all treatments. According to Hafner et al. [50], methanol detected at at this level is considered a minor fermentation end-product when compared with the major VOC-acetic acid and ethanol–in maize silage. However, higher concentrations (583–878 mg/kg DM) can be found in grass silages [51].

Although there are now several studies in the literature that addressed the occurrence, formation, and effects of VOC in silages [30,51], the evidence on their impact on feed intake, milk yield, and animal health is still very limited, and sometimes conflicting results have been reported [2,3]. Acceptance trials with individual VOC [52] are relatively easy to design, but they have not led to consistent results, e.g., for ethanol [53–55]. According to Weiß et al. [13,56], esters in particular, as indicator substances for all VOC, are associated with the occurrence of a large number of volatile compounds and thus an atypical odor. However, the occurrence of these silages with a glue/acetone-like smell is always found in dairy herds worldwide in combination with an impaired feed intake and milk yield. To ensure high feed quality and to avoid any exposure of the animals to potentially toxic or pathogenic substances, the use of silage additives is recommended. Although silage additives containing heterofermentative lactic acid bacteria producing acetic acid can significantly reduce yeast development, they do not always work reliably in terms of suppressing the formation of VOC [18]. Excessive ethanol formation associated with ester formation, as in the present study with contents above 30 g/kg DM, can be suppressed in a reliable manner with chemical ensiling agents containing potassium sorbate, sodium benzoate and/or salts of propionic acid [13,57]. These findings were recently confirmed by da Silva et al. [18], who clearly showed restricted ethanol production, lower counts of lactate-assimilating yeasts and, simultaneously increased ASTA in maize silage stored without and with repeated air ingress and treated with a chemical additive.

## 5. Conclusions

Delayed sealing negatively affects the efficiency of fermentation as reflected by increased DM losses and must be strictly avoided to increase the profitability of silage production. The presence of air during the very early phases of silage storage promotes the activity of undesired yeast activity, changes the fermentation pattern and reduces aerobic stability after silo opening. Prompt sealing can reduce the production of VOC in and emission from silage, including ethyl esters, which have a detrimental effect on the animal and the environment. Repeated air ingress during the later stages of storage after the completion of the main fermentation phase only has a minor effect.

**Author Contributions:** Conceptualization, K.W.; methodology, K.W., H.U.A.; investigation, K.W.; writing—original draft preparation, K.W.; writing—review and editing, K.W., B.K. and H.U.A.; statistical analysis and visualization, B.K.; project administration, K.W. and H.U.A.; funding acquisition, K.W. and H.U.A. All authors have read and agreed to the published version of the manuscript.

**Funding:** This research was funded by the Deutsche Forschungsgemeinschaft (DFG, German Research Foundation)–491192747 and the Open Access Publication Fund of Humboldt-Universität zu Berlin.

**Institutional Review Board Statement:** Not applicable.

**Informed Consent Statement:** Not applicable.

**Data Availability Statement:** Not applicable.

**Acknowledgments:** The article processing charge was funded by the Deutsche Forschungsgemeinschaft (DFG, German Research Foundation)–491192747 and the Open Access Publication Fund of Humboldt-Universität zu Berlin. We deeply acknowledge the expert technical support by the staff of

the Common Laboratory of Analysis, Humboldt Universität zu Berlin, especially by Gabriele Sommer and by Manuela Alt.

**Conflicts of Interest:** The authors declare that there is no conflict of interest.

## Abbreviations

| | |
|---|---|
| DM | Dry matter |
| VOC | Volatile organic compounds |
| EL | Ethyl lactate |
| EA | Ethyl acetate |
| WSC | Water-soluble carbohydrates |

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
