# Peer review of "The Influence of Delayed Sealing and Repeated Air Ingress during the Storage of Maize Silage on Fermentation Patterns, Yeast Development and Aerobic Stability"

_fermentation, doi:10.3390/fermentation8020048_

Round 1
Reviewer 1 Report
The paper describes experiments of a good quality with relevant analyses. a comprehensive study on the effects of delayed sealing and repeated air ingress on the formation of primary fermentation products and other volatile organic compounds (VOC), the development of yeasts and the aerobic stability of maize during the course of fermentation. I suggest that the authors take a look at the following areas:
- What is the purpose of repeated air entry to actual silage production? Why chose this factor?
- It’s better to show the dynamic changes of silage components in the form of figures instead of tables.
- Line 180-182 of part 3.1.2: Please explain the reason of the acetic acid accumulates continuously.
- Lactic acid assimilating yeast is the focus of microbial analysis in this paper. Besides the changes of LAY and total yeast, are lactic acid bacteria affected in different treatments? How is about mold?
- It’s better to place the contents of lines 237-244 on page 9 and lines 256-262 on page 11 in illustrations as explanatory text, there is no need to repeat them in the text.
- The authors should explain why the results (lines 267-268) are contrary to the conclusions in reference [17]?
- In discussion part (line 291-292), the authors mentioned that “Whereas lactic acid formation after 34 days was at the same level (average of all treatments 30.7 g/kg DM) not only independent of air influence but also of delay”. Are there any differences (in color, or smell) in the quality of silage at the end?
Author Response
Please see the answers in the document

Reviewer 2 Report
well researched and written article
I would recommend is to describe better how the silage density was obtained in the bottles and
to address why the density was less than recommended.
Author Response
Please see the answers in the document
